# Tetraose steroidal glycoalkaloids from potato provide resistance against *Alternaria solani* and Colorado potato beetle

Pieter J Wolters*, Doret Wouters, Yury M Tikunov, Shimlal Ayilalath, Linda P Kodde, Miriam F Strijker, Lotte Caarls, Richard GF Visser, Vivianne GAA Vleeshouwers

Wageningen University and Research, Wageningen, Netherlands

**Abstract** Plants with innate disease and pest resistance can contribute to more sustainable agriculture. Natural defence compounds produced by plants have the potential to provide a general protective effect against pathogens and pests, but they are not a primary target in resistance breeding. Here, we identified a wild relative of potato, *Solanum commersonii*, that provides us with unique insight in the role of glycoalkaloids in plant immunity. We cloned two atypical resistance genes that provide resistance to *Alternaria solani* and Colorado potato beetle through the production of tetraose steroidal glycoalkaloids (SGA). Moreover, we provide in vitro evidence to show that these compounds have potential against a range of different (potato pathogenic) fungi. This research links structural variation in SGAs to resistance against potato diseases and pests. Further research on the biosynthesis of plant defence compounds in different tissues, their toxicity, and the mechanisms for detoxification, can aid the effective use of such compounds to improve sustainability of our food production.

*For correspondence:
jaap.wolters@wur.nl

## eLife assessment

This **valuable** study links natural variation in steroidal glycoalkaloid production to disease and insect resistance in potato species. The study design is straightforward and thorough, and the evidence supporting the main conclusions is **solid**. The work will be of interest to plant biologists and breeders.

## Introduction

Worldwide, up to 20–40% of agricultural crop production is lost due to plant diseases and pests (*Savary et al., 2019*). Many crops have become heavily dependent on the use of pesticides, but this is unsustainable as these can negatively affect the environment and their use can lead to development of pesticide resistance (*Calvo-Agudo et al., 2019*; *Hallmann et al., 2014*; *Mikaberidze et al., 2017*; *Lucas et al., 2015*; *Fairchild et al., 2013*; *Landschoot et al., 2017*). The European Union's 'Farm to Fork Strategy' aims to half pesticide use and risk by 2030 (*European Commission Communication from the Commission to the European Parliament, the Council, the European Economic and Social Committee and the Committee of the Regions, 2020*), a massive challenge that illustrates the urgent need for alternative disease control methods.

Wild relatives of crop species are promising sources of natural disease resistance (*Rodewald and Trognitz, 2013*; *Vleeshouwers et al., 2011a*; *Wolters et al., 2019*; *Arora et al., 2019*). Monogenic

**eLife digest** Farmers often rely on pesticides to protect their crops from disease and pests. However, these chemicals are harmful to the environment and more sustainable strategies are needed. This is particularly true for a disease known as the early blight of potato, which is primarily treated using fungicides that stop the fungal pathogen responsible for the infection (*Alternaria solani*) from growing.

An alternative approach is to harness the natural defence systems that plants already have in place to protect themselves. Like humans, plants have an immune system which can detect and destroy specific pathogens. On top of this, they release defence compounds that are generally toxic to pests and microbes, stopping them from infiltrating and causing an infection.

In 2021, a group of researchers discovered a wild relative of the potato, known as *Solanum commersonii,* with strong resistance to early blight disease. Here, Wolters et al. – including some of the researchers involved in the 2021 study – set out to find how this plant defends itself from the fungus *A. solani.*

The team found that two closely linked genes are responsible for the resistant behaviour of *S. commersonii,* which both encode enzymes known as glycosyltransferases. Further experiments revealed that the enzymes protect *S. commersonii* from early blight disease by modifying steroidal glycoalkaloids, typical defence compounds found in potato and other plants from the same family. The glycosyltransferases alter glycoalkaloids in *S. commersonii* by adding a sugar group to a specific part of the compound called glycone.

Wolters et al. found that the glycoalkaloids from *S. commersonii* were able to slow the growth of other fungal pathogens that harm potatoes when tested in the laboratory. They also made plants resistant to another common destroyer of crops, the Colorado potato beetle.

These findings could help farmers breed potatoes and other crops that are more resistant to early blight disease and Colorado potato beetle, as well as potentially other fungi and pests. However, further experiments are needed to investigate how these glycone-modified glycoalkaloids affect humans, and how variants of glycoalkaloids are produced and degraded in different parts of the plants. Acquiring this knowledge will help to employ these defence compounds in a safe and effective manner.

---

resistance caused by dominant resistance (*R*) genes, typically caused by immune receptors that belong to the nucleotide-binding leucine-rich repeat (NLR) class, is successfully employed by plant breeders to develop varieties with strong qualitative disease resistance. However, this type of resistance is usually restricted to a limited range of pathogens (*Flor, 1971*; *Jones and Dangl, 2006*) and it is often not very durable.

More robust resistance can be obtained by combining NLRs with different recognition specificities (*Kim et al., 2012*; *Zhu et al., 2012*; *Vleeshouwers et al., 2011b*; *Rietman et al., 2012*), or by including pattern recognition receptors (PRRs), which recognise conserved (microbe- or pathogen-derived) molecular patterns. Recent reports show that PRRs and NLRs cooperate to provide disease resistance (*Ngou et al., 2021*; *Yuan et al., 2021*; *Rhodes et al., 2022*). Alternatively, susceptibility (*S*) genes provide recessive resistance that can be both broad-spectrum and durable (*van Schie and Takken, 2014*; *Jørgensen, 1992*; *Sun et al., 2022*). Unfortunately, their recessive nature complicates the use of *S* genes in conventional breeding of autopolyploids and many mutated *S* genes come with pleiotropic effects.

Besides the defences mentioned above, most plants produce specialised metabolites with antimicrobial or anti-insect activity, either constitutively (phytoanticipins) or in response to pathogen attack or herbivory (phytoalexins) (*VanEtten et al., 1994*). These natural defence compounds are derived from a wide range of building blocks, leading to a large structural diversity throughout the plant kingdom (*Piasecka et al., 2015*; *Dixon, 2001*). Specific classes of compounds can be found in different plant families, for example glucosinolates are typically found in Brassicaceae and benzoxazinoids are widely distributed among Poaceae, with further chemical diversification within plant families (*de Bruijn et al., 2018*; *Halkier and Gershenzon, 2006*). Examples in various pathosystems show that the capacity for detoxification of plant defence compounds is important for pathogenicity, especially

for necrotrophic pathogens which encounter toxic plant metabolites as a consequence of their life-style (*VanEtten et al., 1995*; *Westrick et al., 2021*). Similarly, plant defence compounds play an important role against herbivorous insects, which have evolved various mechanisms to deal with toxic plant compounds (*Després et al., 2007*; *Heidel-Fischer and Vogel, 2015*; *Calla et al., 2017*).

While plant immune receptors can offer resistance against a restricted range of pathogens, plant defence compounds have the potential to provide a more general protection, depending on their mode of action. Plants from the nightshade family (Solanaceae) produce saponins that are character-ised by a steroidal alkaloid aglycone, linked to a variable oligosaccharide chain (steroidal glycoalka-loids – SGAs) (*Li et al., 2006*; *Heftmann, 1983*). The protective effect of SGAs stems from their ability to interact with membrane sterols, disrupting the cell integrity from target organisms (*Keukens et al., 1995*; *Armah et al., 1999*; *Fenwick et al., 1991*; *Osbourn, 1996*). In addition, they can act on the nervous system of pests and herbivores through their inhibitory effect on cholinesterase enzymes (*Orgell et al., 1958*; *Wierenga and Hollingworth, 1992*; *Roddick, 1996*). As a consequence, SGAs can have both antimicrobial and anti-insect activity (*Chowański et al., 2016*; *Munafo and Gianfagna, 2011*; *You and van Kan, 2021*; *Sinden et al., 1986*; *Sinden et al., 1980*; *Tai et al., 2015*; *Tai et al., 2014*; *Paudel et al., 2019*; *Kaup et al., 2005*; *Seipke and Loria, 2008*; *Paudel et al., 2017*).

Early blight is an important disease of tomato and potato that is caused by the necrotrophic fungal pathogen *Alternaria solani* (*Christ, 1989*; *Rotem, 1994*; *Shtienberg et al., 1990*). In a previous study, we found a wild potato species, *Solanum malmeanum* (also referred to as *S. commersonii* subsp. *malmeanum* [*Spooner et al., 2014*]), with strong resistance against potato pathogenic *Alternaria* isolates and species from a number of different locations (*Wolters et al., 2021*). We showed that resistance is likely caused by a single dominant locus and that it can be introgressed in cultivated potato (*Wolters et al., 2021*). Resistance to necrotrophs is usually considered to be a complex, poly-genic trait, or recessively inherited according to the *inverse gene-for-gene* model (*Glazebrook, 2005*; *Vleeshouwers and Oliver, 2014*; *Lorang et al., 2007*; *Nagy and Bennetzen, 2008*; *Faris et al., 2010*; *Shi et al., 2016*). It therefore surprised us to find a qualitative dominant resistance against early blight in *S. commersonii* (*Wolters et al., 2021*).

In this study, we explored different accessions of *S. commersonii* and *S. malmeanum* and devel-oped a population that segregates for resistance to early blight. Using a bulked segregant RNA-Seq (BSR-Seq) approach (*Dobnik et al., 2021*), we mapped the resistance locus to the top of chromosome 12 of potato. We sequenced the genome of the resistant parent of the population and identified two glycosyltransferases (GTs) that can provide resistance to susceptible *S. commersonii*. We show that the resistance is based on the production of tetraose SGAs and provide in vitro evidence to show that they can be effective against other fungi besides *A. solani*. As SGAs can be involved in resistance against insects, we also tested if they can protect against Colorado potato beetle (CPB). Combined, our results show that the tetraose SGAs from *S. commersonii* have potential to provide resistance against a range of potato pathogens and pests.

## Results
### Early blight resistance maps to chromosome 12 of potato

To find suitable parents for a mapping study targeting early blight resistance, we performed a disease screen with *A. solani* isolate altNL03003 (*Wolters et al., 2018*) on 13 different accessions encompassing 37 genotypes of *S. commersonii* and *S. malmeanum* (*Supplementary file 1*). The screen showed clear differences in resistance phenotypes between and within accessions (*Figure 1a*). Roughly half of the genotypes were highly resistant (lesion diameters <3 mm indicate that the lesions are not expanding beyond the size of the inoculation droplet) and the other half was susceptible (displaying expanding lesions), with only a few intermediate genotypes. CGN18024 is an example of an accession that segre-gates for resistance, with CGN18024_1 showing strong resistance and CGN18024_3 showing clear susceptibility (*Figure 1b*). The fact that individual accessions can display such clear segregation for resistance suggests that resistance is caused by a single gene or locus. Because of its clear segrega-tion, *S. commersonii* accession CGN18024 was selected for further studies.

Disease tests with an *A. solani* isolate from the US (ConR1H) and a recent Dutch isolate from the Netherlands (altNL21001) confirm that the resistance of CGN18024_1 is effective against additional *A. solani* isolates (*Figure 1—figure supplement 1*). To further study the genetics underlying resistance to

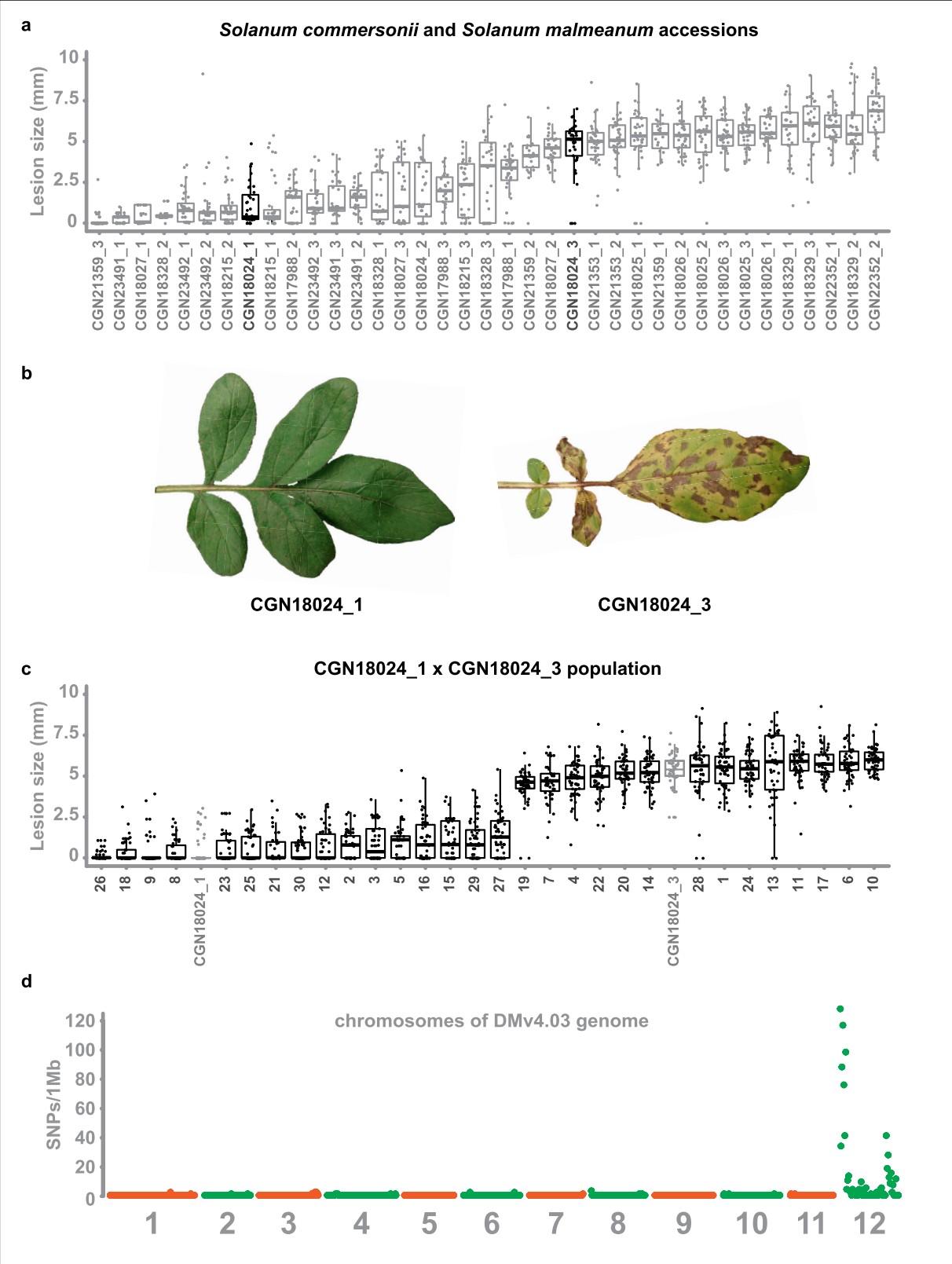

**Figure 1.** Early blight resistance maps to chromosome 12 of potato. (**a**) Two to three genotypes of 13 different accessions of *S. commersonii* and *S. malmeanum* were inoculated with *A. solani* altNL03003. Three plants of each genotype were tested and three leaves per plants were inoculated with six 10 µl droplets with spore suspension. Lesion diameters were measured 5 days post inoculation and visualised using boxplots, with horizonal lines indicating median values and individual measurements plotted on top. Non-expanding lesions (<3 mm) indicate resistance and expanding lesions

*Figure 1 continued on next page*

*Figure 1 continued*

indicate susceptibility. Some accessions segregate for resistance. (**b**) Accession CGN18024 is an example of an accession that segregates for resistance to *A. solani*, with CGN18024_1 displaying resistance and CGN18024_3 displaying susceptibility at 5 days after spray-inoculation. (**c**) Progeny from CGN18024_1 × CGN18024_3 was inoculated with *A. solani*. Three plants of each genotype were tested and three leaves per plants were inoculated with six 10 µl droplets with spore suspension each. Lesion diameters were measured 5 days post inoculation. 16 progeny genotypes are resistant (with lesion diameters <2–3 mm) and 14 are susceptible (with expanding lesions). This corresponds to a 1:1 segregation ratio ($X^2$ (1, $N$ = 30) = 0.133, $\rho$ = 0.72). (**d**) SNPs derived from a BSR-Seq analysis (heterozygous in resistant parent CGN18024_1 and the resistant bulk, but absent or homozygous in susceptible parent and susceptible bulk) using bulks of susceptible and resistant progeny were plotted in 1 Mb windows over the 12 chromosomes of the potato DMv4.03 genome (*Xu et al., 2011*). They are almost exclusively located on chromosome 12.

The online version of this article includes the following source data and figure supplement(s) for figure 1:

**Source data 1.** Numerical data underlying *Figure 1a*.

**Source data 2.** Numerical data underlying *Figure 1c*.

**Source data 3.** Numerical data underlying *Figure 1d*.

**Figure supplement 1.** Resistance of *S commersonii* genotypes CGN18024_1 and CGN18024_3 against different isolates of *A.solani*.

**Figure supplement 2.** Resistance from *S.commersonii* to *A. solani* is mapped to the top of chromosome 12.

early blight, we crossed resistant CGN18024_1 with susceptible CGN18024_3. Thirty progeny genotypes were sown out and tested with *A. solani* isolate altNL03003. We identified 14 susceptible genotypes and 16 resistant genotypes, with no intermediate phenotypes in the population (*Figure 1c*). This segregation supports a 1:1 ratio ($X^2$ (1, $N$ = 30) = 0.133, p = 0.72), which confirms that resistance to early blight is likely caused by a single dominant locus in *S. commersonii*.

To genetically localise the resistance, we isolated RNA from each progeny genotype and the parents of the population and proceeded with a BSR-Seq analysis (*Dobnik et al., 2021*). RNA from resistant and susceptible progeny genotypes were pooled in separate bulks and sequenced next to RNA from the parents on the Illumina sequencing platform (PE150). Reads were mapped to the DMv4.03 (*Xu et al., 2011*) and Solyntus potato genomes (*Hoopes et al., 2022*). To find putative SNPs linked to resistance, we filtered for SNPs that follow the same segregation as resistance (heterozygous in resistant parent CGN18024_1 and the resistant bulk, but absent or homozygous in susceptible parent and susceptible bulk). The resulting SNPs localise almost exclusively on chromosome 12 of the DM and Solyntus genomes, with most of them located at the top of the chromosome (*Figure 1d*). We used a selection of SNPs distributed over chromosome 12 as high-resolution melt (HRM) markers to genotype the BSR-Seq population. This rough mapping proves that the locus for early blight resistance resides in a region of 3 Mb at the top of chromosome 12 (*Figure 1—figure supplement 2*).

## Improved genome assembly *of S. commersonii*

A genome sequence of *S. commersonii* is already available (*Aversano et al., 2015*), but we do not know if the sequenced genotype is resistant to *A. solani*. To help the identification of additional markers and to explore the resistance locus of a genotype with confirmed resistance, we sequenced the genome of resistant parent CGN18024_1. High-molecular-weight genomic DNA from CGN18024_1 was used for sequencing using Oxford Nanopore Technology (ONT) on a GridION X5 platform and for sequencing using DNA Nanoball (DNB) technology at the Beijing Genomics Institute (BGI) to a depth of approximately 30×. We used the ONT reads for the initial assembly and the shorter, more accurate, DNBseq reads to polish the final sequence. The resulting assembly has a size of 737 Mb, which is close to the size of the previously published genome of *S. commersonii* (730 Mb) (*Aversano et al., 2015*). N50 scores and Benchmarking Universal Single-Copy Orthologs

**Table 1.** Genome assembly metrics of *S. commersonii* cmm1t (*Aversano et al., 2015*) and CGN18024_1.

| Genome | CMM1t* | CGN18024_1 |
|---|---|---|
| Total size (Mb) | 730 | 737 |
| Contig number | 278,460 | 637 |
| Largest contig (Mb) | 0.17 | 21.2 |
| N50 (Mb) | 0.007 | 4.02 |
| Complete BUSCO (%) | 81.9 | 95.7 |

*\*Aversano et al., 2015*.

(BUSCO) score indicate a highly complete and contiguous genome assembly of *S. commersonii* (*Table 1*).

## Identification of two GT resistance genes

To identify candidate genes that can explain the resistance of *S. commersonii*, it was necessary to further reduce the mapping interval. By aligning the ONT reads to the CGN18024_1 genome assembly, we could identify new polymorphisms that we converted to additional PCR markers (*Figure 2—figure supplements 1–4*). We performed a recombinant screen of approximately 3000 genotypes from the population to fine map the resistance region to a window of 20 kb (*Figure 2—figure supplements 5 and 6*).

We inferred that the resistance locus is heterozygous in CGN18024_1 from the segregation in the mapping population. We used polymorphisms in the resistance region to separate and compare the ONT sequencing reads from the resistant and susceptible haplotype. This comparison showed a major difference between the two haplotypes. The susceptible haplotype contains a small insertion of 3.7 kb inside a larger region of 7.3 kb. The larger region is duplicated in the resistant haplotype (*Figure 2a*). As a result, the resistance region of the resistant haplotype is 27 kb, 7 kb larger than the corresponding region of the susceptible haplotype (20 kb).

Two genes coding for putative GTs are located within the rearrangement of the resistant haplotype. The corresponding allele from the susceptible haplotype contains a frameshift mutation, leading to a truncated protein (*Figure 2—figure supplement 7*). Several other short ORFs with homology to GTs were predicted in the resistant haplotype, but *ScGTR1* (*S. commersonii* GT linked to resistance 1) and *ScGTR2* are the only full-length genes in the region. Reads from the BSR-Seq experiment show that both genes are expressed in bulks of resistant progeny and not in susceptible progeny (*Figure 2b*), suggesting a putative role for these genes in causing resistance. *ScGTR1* and *ScGTR2* are homologous genes with a high similarity (the predicted proteins that they encode share 97% amino acid identity). We compared the predicted amino acid sequences with previously characterised GTs (*Bowles et al., 2005*; *McCue et al., 2007*; *McCue et al., 2006*; *McCue et al., 2005*; *Masada et al., 2009*; *Itkin et al., 2013*; *Itkin et al., 2011*; *Tikunov et al., 2013*) and found that they share some similarity with GTs with a role in zeatin biosynthesis (*Martin et al., 1999a*; *Martin et al., 1999b*; *Mok et al., 2005*) and with GAME17, an enzyme involved in biosynthesis of the SGA α-tomatine typically found in tomato (*Itkin et al., 2013*; *Figure 2—figure supplement 8*, *Supplementary file 2*).

To test whether the identified candidate genes are indeed involved in resistance, we transiently expressed both alleles of the resistant haplotype (*ScGTR1* and *ScGTR2*) as well as the corresponding allele from the susceptible haplotype (*ScGTS*), in leaves of resistant CGN18024_1 and susceptible CGN18024_3 and *S. tuberosum* cultivar Atlantic, using agroinfiltration (*Lazo et al., 1991*). Following agroinfiltration, the infiltrated areas were drop inoculated with a spore suspension of *A. solani* altNL03003. Transient expression of *ScGTR1* as well as *ScGTR2* significantly reduced the size of the *A. solani* lesions in susceptible CGN18024_3, compared to *ScGTS* and the empty vector control. Resistant CGN18024_1 remained resistant, whereas susceptible Atlantic remained susceptible regardless of the treatment (*Figure 2c*). We conclude that both *ScGTR1* and *ScGTR2* can affect resistance in susceptible *S. commersonii* CGN18024_3, but not in *S. tuberosum* cv. Atlantic.

## Leaf compounds from resistant *S. commersonii* inhibit growth of diverse fungi, including pathogens of potato

GTs are ubiquitous enzymes that catalyse the transfer of saccharides to a range of different substrates. To test if resistance of *S. commersonii* to *A. solani* can be explained by a host-specific defence compound, we performed a growth inhibition assay using crude leaf extract from resistant and susceptible *S. commersonii*. Leaf material was added to PDA plates to equal 5% (wt/vol) and autoclaved (at 121°C) or semi-sterilised at 60°C. Interestingly, leaf material from resistant CGN18024_1 strongly inhibited growth of *A. solani* isolate altNL03003, while we did not see any growth inhibition on plates containing leaves from susceptible CGN18024_3 (*Figure 3a*). Remarkably, ample contamination with diverse fungi appeared after a few days on the plates containing semi-sterilised leaves from susceptible *S. commersonii* but not on plates with leaves from CGN18024_1 (*Figure 3a*). Thus, leaves from CGN18024_1 contain compounds that can inhibit growth of a variety of fungi besides *A. solani*.

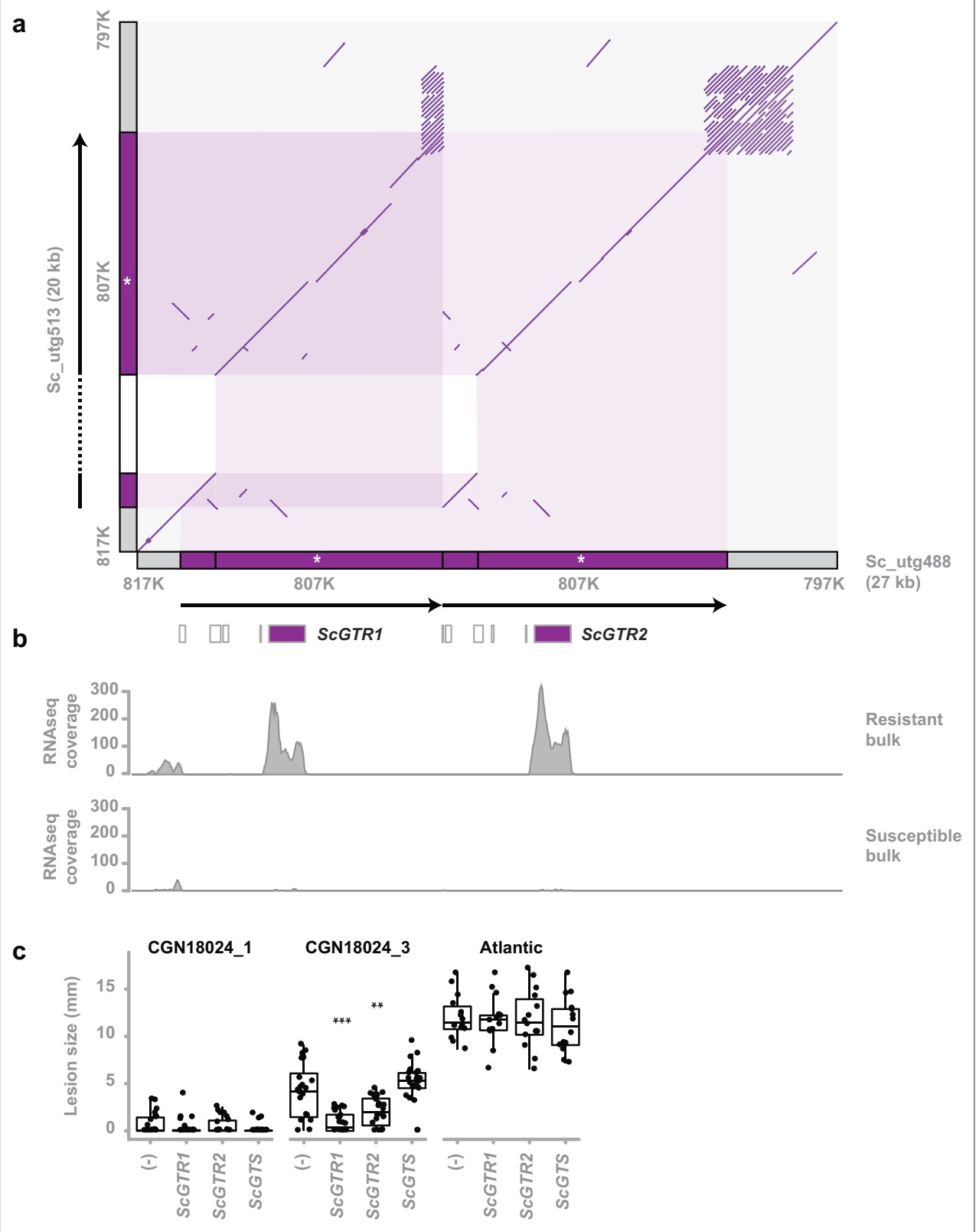

**Figure 2.** Identification of two glycosyltransferase resistance genes. (**a**) Comparison of the susceptible and resistant haplotype of the *Solanum commersonii* CGN18024_1 resistance region (delimited by markers 817K and 797K) in a comparative dot plot shows a rearrangement. Locations of markers used to map the resistance region are indicated in grey along the *x*- and *y*-axis. The duplicated region of the resistant haplotype contains marker 807K (white asterisk) and two predicted glycosyltransferase genes (*ScGTR1* and *ScGTR2*). Several short ORFs with homology to

*Figure 2 continued on next page*

*Figure 2 continued*

glycosyltransferase genes that were predicted in the resistance region are indicated by white boxes, but *ScGTR1* and *ScGTR2* are the only full-length genes. As a result of the rearrangement, the resistance region of the resistant haplotype (27 kb) is 7 kb larger than the corresponding region of the susceptible haplotype (20 kb). (b) Alignment of RNAseq reads from the BSR-Seq analysis shows that *ScGTR1* and *ScGTR2* are expressed in bulks of resistant progeny, but not in bulks of susceptible progeny. (c) *S. tuberosum* cv. 'Atlantic', *S. commersonii* CGN18024_1 and CGN18024_3 were agroinfiltrated with expression constructs for *ScGTR1* and *ScGTR2*, *ScGTS*, and empty vector (-), combined as seperate spots on the three middle leaves of each genotype (8 plants per genotype). *A. solani* is inoculated in the middle of agroinfiltrated areas at 2 days after agroinfiltration and lesion diameters are measured 5 days after inoculation. Lesion sizes were visualised with boxplots, with horizonal lines indicating median values and individual measurements plotted on top. Agroinfiltration with expression constructs for *ScGTR1* and *ScGTR2* results in a significant (Welch's two-sample *t*-test, **p < 0.01, ***p < 0.001) reduction of lesion sizes produced by *Alternaria solani* altNL03003 in *S. commersonii* CGN18024_3, but not in *S. tuberosum* cv. 'Atlantic'.

The online version of this article includes the following source data and figure supplement(s) for figure 2:

**Source data 1.** Numerical data underlying *Figure 2c*.

**Figure supplement 1.** Overview of marker 817K.

**Figure supplement 2.** Overview of marker 807K.

**Figure supplement 3.** Overview of marker 797K.

**Figure supplement 4.** Overview of marker 764K.

**Figure supplement 5.** Fine mapping the resistance locus in CGN18024_1.

**Figure supplement 6.** Early blight disease symptoms on key recombinants.

**Figure supplement 7.** Alignment of putative *S. commersonii* glycosyltransferases (ScGTs) linked to resistance.

**Figure supplement 8.** Comparative phylogenetic analysis of glycosyltransferases with a known function (*Supplementary file 2*).

To quantify the inhibitory effect of leaves from *S. commersonii* against different fungal pathogens of potato, we performed a growth inhibition assay with *A. solani* (altNL03003 [*Wolters et al., 2018*]), *Botrytis cinerea* (B05.10 [*Amselem et al., 2011*]), and *Fusarium solani* (1992 vr). As before, we added 5% (wt/vol) of leaf material from CGN18024_1 or CGN18024_3 to PDA plates and we placed the fungi at the centre of the plates. We measured colony diameters in the following days and compared it with the growth on PDA plates without leaf extract. Indeed, growth of all three potato pathogens was significantly reduced on medium containing leaf material from CGN18024_1 (*Figure 3b*), compared to medium containing material from CGN18024_3 or on normal PDA plates. These results indicate that phytoanticipins from the leaves of resistant *S. commersonii* have the potential to protect against diverse fungal pathogens of potato.

## Tetraose SGAs from *S. commersonii* provide resistance to *A. solani* and CPB

*Solanum* leaves usually contain SGAs, which are known phytoanticipins against fungi and other plant parasites (*Friedman, 2006*). *S. tuberosum* typically produces the triose SGAs α-solanine (solanidine-Gal-Glu-Rha) and α-chaconine (solanidine-Glu-Rha-Rha), while five major tetraose SGAs were previously identified in *S. commersonii*: commersonine (demissidine-Gal-Glu-Glu-Glu), dehydrocommersonine (solanidine-Gal-Glu-Glu-Glu), demissine (demissidine-Gal-Glu-Glu-Xyl), dehydrodemissine (solanidine-Gal-Glu-Glu-Xyl), and α-tomatine (tomatidine-Gal-Glu-Glu-Xyl) (*Friedman, 2006*; *Osman et al., 1976*; *Friedman et al., 1997*; *Distl and Wink, 2009*; *Caruso et al., 2011*; *Vázquez et al., 1997*). To test if SGAs can explain resistance of *S. commersonii*, we measured SGA content in leaves from Atlantic and susceptible/resistant *S. commersonii* by ultra high-performance liquid chromatography (UPLC) coupled to mass spectrometry (MS). As expected, we could detect the triose SGAs α-solanine and α-chaconine in susceptible *S. tuberosum* cv. Atlantic, but we found a remarkable difference in the SGA profile of resistant and susceptible *S. commersonii*. We detected tetraose SGAs in resistant *S. commersonii* CGN18024_1, whereas susceptible *S. commersonii* CGN18024_3 accumulates triose SGAs (*Figure 4a* and *Supplementary files 3 and 4*). The mass spectra of the four major tetraose SGAs from *S. commersonii* correspond to (dehydro-) commersonine and (dehydro-) demissine, matching the data from previous studies (*Osman et al., 1976*; *Distl and Wink, 2009*; *Caruso et al., 2011*; *Vázquez et al., 1997*). Notably, the mass spectra of the two major SGAs from susceptible CGN18024_3 correspond to the triose precursors of these SGAs (solanidine-Gal-Glu-Glu and demissidine-Gal-Glu-Glu, respectively) (*Supplementary files 3 and 4*). These results suggest that the triose SGAs present in

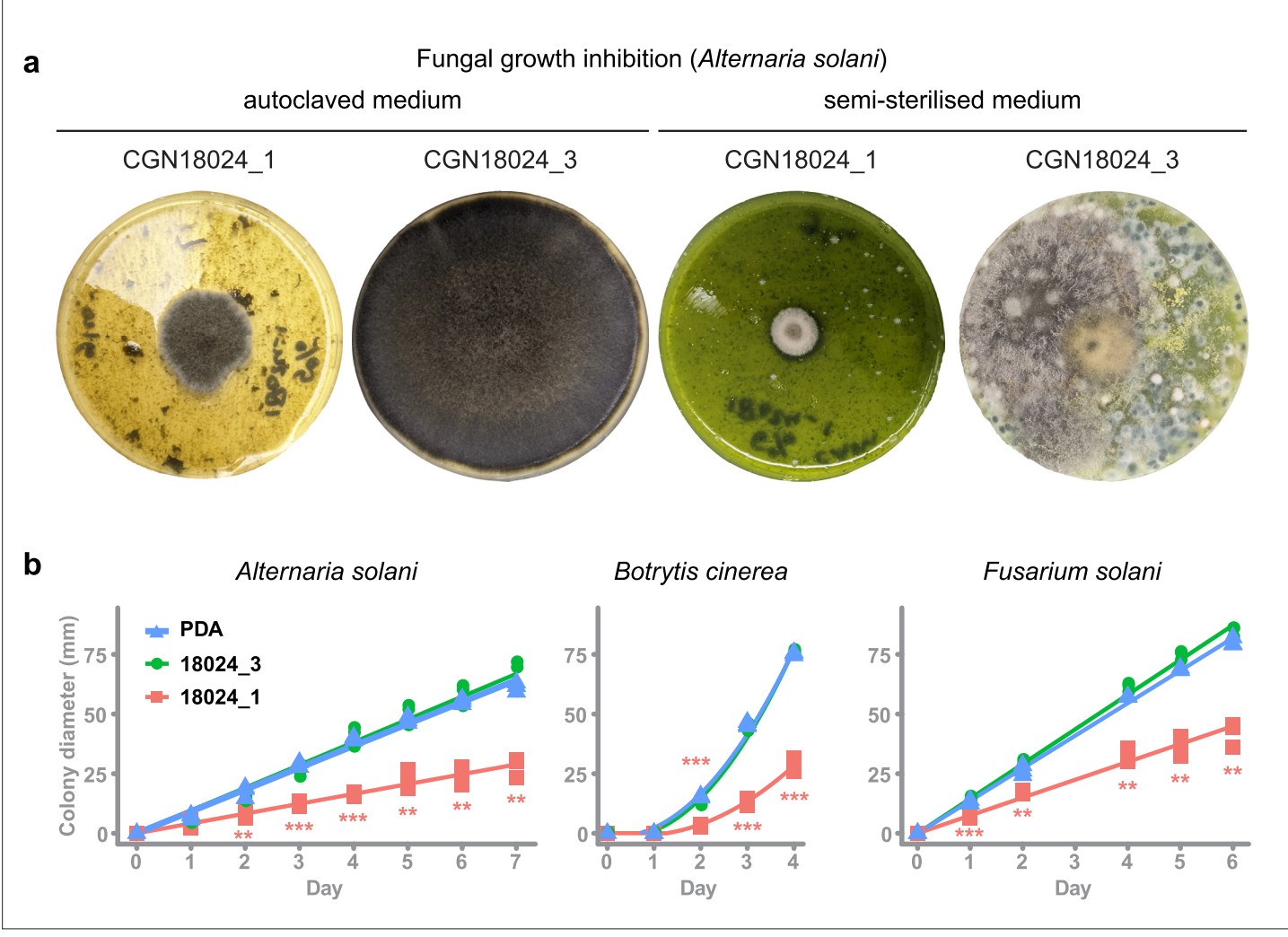

**Figure 3.** Leaf compounds from resistant *S. commersonii* inhibit growth of diverse fungi, including pathogens of potato. (**a**) Crude leaf extract from CGN18024_1/CGN18024_3 was added to PDA plates (5%, wt/vol) and autoclaved (left) or semi-sterilised for 15 min at 60°C (right). Growth of *Alternaria solani* altNL03003 was strongly inhibited on PDA plates with autoclaved leaf extract from CGN18024_1 compared to plates with CGN18024_3, as shown on the left two pictures taken at 7 days after placing an agar plug with mycelium of *A. solani* at the centre of each plate. Abundant fungal contamination appeared after 4 days on plates containing semi-sterilised leaf from CGN18024_3, but not on plates containing material from CGN18024_1 (right two pictures). (**b**) Growth of potato pathogenic fungi *A. solani* (altNL03003), *B. cinerea* (B05.10), and *F. solani* (1992 vr) was followed by measuring the colony diameter on PDA plates containing autoclaved leaf material from CGN18024_1/CGN18024_3. Growth of all three fungi was measured on PDA plates containing CGN18024_1 (red squares), CGN18024_3 (green circles), or plates with PDA and no leaf material (blue triangles). Three replicates were used per isolate/substrate combination. Significant differences in growth on PDA plates containing plant extract compared to PDA plates without leaf extract are indicated with asterisks (Welch's two-sample *t*-test, **p < 0.01, ***p < 0.001).

The online version of this article includes the following source data for figure 3:

**Source data 1.** Numerical data underlying *Figure 3b*.

susceptible CGN18024_3 are modified to produce the tetraose SGAs in resistant CGN18024_1, by addition of an extra glucose or xylose moiety.

To investigate a possible role for *ScGTR1* and *ScGTR2* in the production of tetraose SGAs from CGN18024_1 and their link to resistance, we generated stable transformants of *ScGTR1* and *ScGTR2* in triose SGA producing CGN18024_3 (*Figure 4—figure supplement 1*). UPLC–MS analysis showed that both *ScGTR1* and *ScGTR2* transformants accumulate tetraose SGAs, while the amount of triose SGAs is markedly reduced (*Figure 4a*). Strikingly, ScGTR1 and ScGTR2 appear to have different specificities. Overexpression of *ScGTR1* resulted in the addition of a hexose to the triose SGAs from CGN18024_3 (corresponding to a commertetraose), while overexpression of *ScGTR2* caused

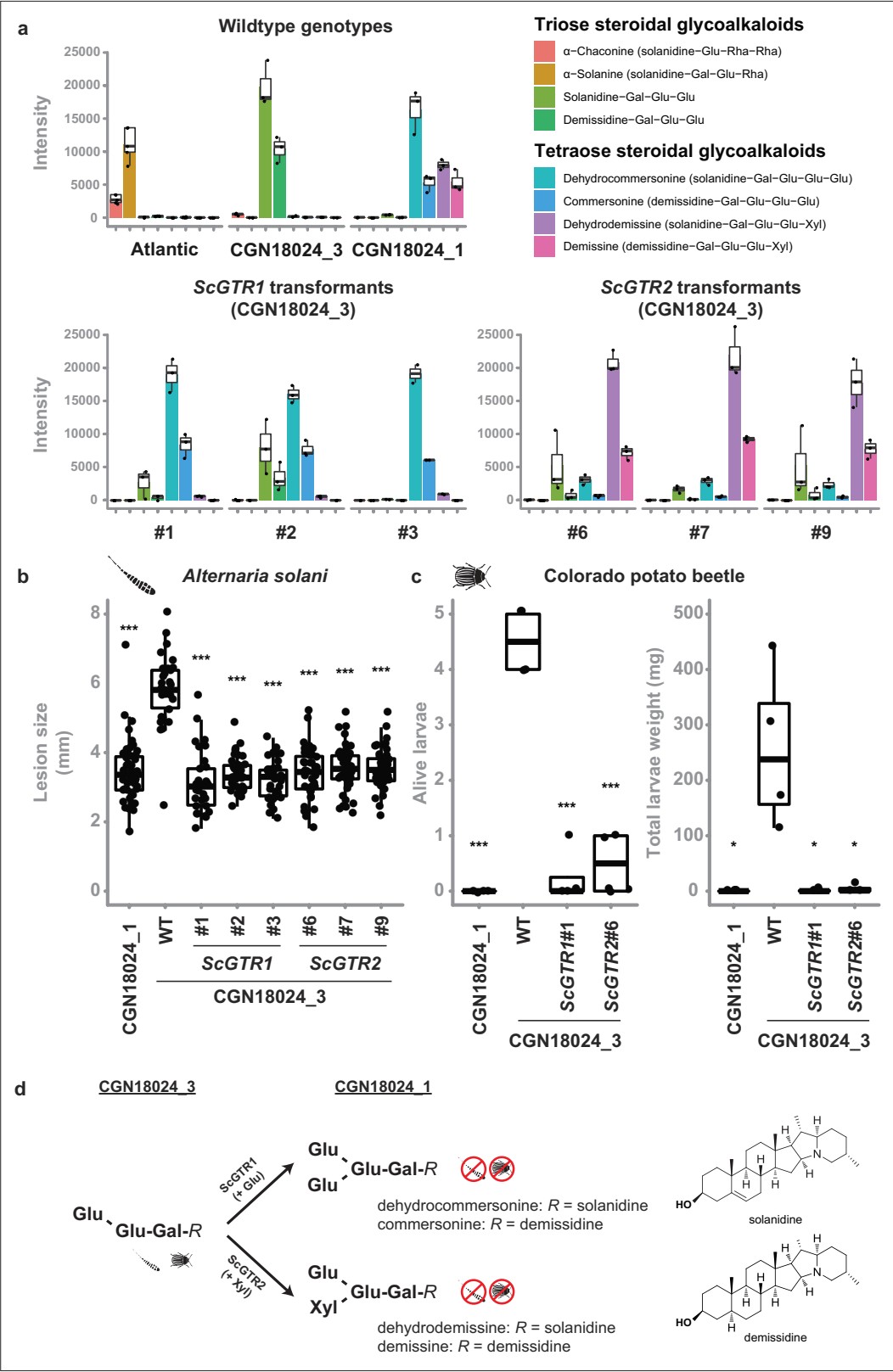

**Figure 4.** Tetraose steroidal glycoalkaloids (SGAs) from *Solanum commersonii* provide resistance to *Alternaria solani* and Colorado potato beetle. Data are visualised with boxplots, with horizonal lines indicating median values and individual measurements plotted on top. (**a**) Tetraose SGAs were detected in resistant CGN18024_1 and in CGN18024_3 transformed with *ScGTR1/ScGTR2*. Susceptible *S. tuberosum* cv. 'Atlantic' and wildtype (WT)

*Figure 4 continued on next page*

*Figure 4 continued*

CGN18024_3 contain only triose SGAs. Overexpression of *ScGTR1* resulted in the addition of a hexose to the triose SGAs from CGN18024_3, resulting in a commertetraose (Gal-Glu-Glu-Glu), while overexpression of *ScGTR2* caused the addition of a pentose, resulting in a lycotetraose (Gal-Glu-Glu-Xyl). Each boxplot displays the data of three seperate measurements (**b**) WT CGN18024_1/CGN18024_3 and CGN18024_3 transformants were inoculated with *Alternaria solani* altNL03003. Three plants of each genotype were tested and three leaves per plants were inoculated with six 10 µl droplets with spore suspension each. Lesions diameters were measured 5 days post inoculation. Significant differences with WT CGN18024_3 are indicated with asterisks (Welch's two-sample *t*-test, ***p < 0.001). *ScGTR1* and *ScGTR2* can both complement resistance to *A. solani* in CGN18024_3, as the lesion sizes produced on CGN18024_3 transformants are comparable to resistant CGN18024_1. (**c**) Three plants per genotype were challenged with five Colorado potato beetle larvae each. The tetraose SGAs produced by ScGTR1 and ScGTR2 can provide resistance to Colorado potato beetle, as indicated by reduced larvae survival and total larvae weight. Significant differences with WT CGN18024_3 are indicated with asterisks (Welch's two-sample *t*-test, *p < 0.05, ***p < 0.001). (**d**) Putative structures of SGAs detected in CGN18024_1 and CGN18024_3, based on previous studies (*Osman et al., 1976*; *Distl and Wink, 2009*; *Caruso et al., 2011*; *Vázquez et al., 1997*). CGN18024_3 produces triose SGAs and is susceptible to Colorado potato beetle and *A. solani*. ScGTR1 and ScGTR2 from CGN18024_1 convert these triose SGAs from susceptible *S. commersonii* to tetraose SGAs, through the addition of a glucose or xylose moiety, respectively. Both sugar additions can provide resistance to Colorado potato beetle and *A. solani*.

The online version of this article includes the following source data and figure supplement(s) for figure 4:

**Source data 1.** Numerical data underlying *Figure 4a*.

**Source data 2.** Numerical data underlying *Figure 4b*.

**Source data 3.** Numerical data underlying *Figure 4c*.

**Figure supplement 1.** Validation of *ScGTR1* and *ScGTR2* transformants using PCR.

**Figure supplement 2.** Principal component analysis (PCA) on *Solanum commersonii* genotypes and transformants.

the addition of a pentose (corresponding to a lycotetraose) (*Figure 4a, d*). This in planta evidence suggests that ScGTR1 is a glucosyltransferase and that ScGTR2 is a xylosyltransferase. However, we detect a slight overlap in activity. In addition to the lycotetraose products, we detected small amounts of commertetraose product in *ScGTR1* transformants and vice versa in the *ScGTR2* transformants (*Figure 4a* and *Supplementary file 3*). A multivariate principal components analysis (PCA) on the full metabolic profile consisting of all 1041 detected mass peaks revealed that ScGTR1 and ScGTR2 are highly specific towards SGAs since 75% of the metabolic variation between the transformants and the wild types could be explained by the SGA modifications (*Figure 4—figure supplement 2*). Modifications catalysed by both enzymes can lead to resistance, as *ScGTR1* and *ScGTR2* transformants are both resistant to *A. solani* isolate altNL03003 (*Figure 4b*). Atlantic *ScGTR1* and *ScGTR2* transformants did not show differences in SGA profile, probably because they contain different triose SGA substrates than found in *S. commersonii* CGN18024_3 (*Supplementary files 3 and 4*).

Leptine and dehydrocommersonine SGAs from wild potato relatives have previously been linked to resistance to insects such as CPB (*Chowański et al., 2016*; *Sinden et al., 1986*; *Sinden et al., 1980*; *Tai et al., 2015*; *Tai et al., 2014*; *Paudel et al., 2019*; *Sagredo et al., 2009*). To see if the SGAs from *S. commersonii* can protect against insects as well, we performed a test with larvae of a CPB genotype collected in the Netherlands on wildtype CGN18024_1/CGN18024_3 and on CGN18024_3 transformed with *ScGTR1* or *ScGTR2* (*Figure 4b*). Wildtype CGN18024_3 is susceptible to the CPB genotype that was tested, but CGN18024_1 and CGN18024_3 transformed with *ScGTR1* or *ScGTR2* are resistant, as illustrated by a very low larvae weight and survival (*Figure 4c*). Thus, the conversion of triose SGAs from CGN18024_3 to tetraose SGAs produced by CGN18024_1, carried out by both ScGTR1 and ScGTR2, can provide protection against *A. solani* as well as CPB (*Figure 4a–d*).

## Discussion

In this study, we set out to characterise resistance of *S. commersonii* to *A. solani*. We showed that it is caused by a single dominant locus containing two GT candidate resistance genes. Both GTs are involved in the production of tetraose SGAs in *S. commersonii*, but they transfer distinct sugars. Both

modifications can cause resistance to *A. solani*. We provide in vitro evidence to show that the tetraose SGAs from *S. commersonii* have the potential to protect against other fungi besides *A. solani* and we demonstrate that plants producing the compounds are resistant to CPB. Collectively, our data link the tetraose SGAs from *S. commersonii* to disease and pest resistance.

It is known that specialised metabolites from plants can act in plant defence and compounds with antimicrobial effects have been characterised in many different plant species (*Piasecka et al., 2015*; *Dixon, 2001*; *Polturak and Osbourn, 2021*). They may also influence other aspects of the crop, such as flavour or taste and they can have dietary benefits or be toxic to humans. SGAs from potato can cause risks for human health, but a total SGA content of less than 200 mg/kg is generally considered to be safe for human consumption (*Friedman, 2006*; *Valkonen et al., 1996*; *Schrenk et al., 2020*; *Dolan et al., 2010*). Potato breeders generally try to reduce SGA content in tubers, to prevent problems with toxicity and to meet safety regulations, but they do not usually consider the effect on disease resistance. There is not much known about how modifications to SGAs of potato affect human toxicity and resistance to biotic stress, but additional knowledge on this topic could help breeders to optimise the metabolite profile of their cultivars (*Baur et al., 2022*).

Biosynthesis of SGAs in *Solanum* is controlled by many genes. The discovery of *S. commersonii* genotypes with and without tetraose SGAs provides us with unique insight in the role of these compounds in plant immunity. Similar compounds are produced in *Solanum* species such as *S. chacoense*, *S. chomatophilum*, *S. oplocense*, *S. paucisectum*, and *S. piurae*, which may explain why these (or their descendants) display resistance to *A. solani* or CPB (*Sinden et al., 1980*; *Tai et al., 2015*; *Tai et al., 2014*; *Ding et al., 2019*; *Alam, 1985*). The compounds that are found in resistant *S. commersonii* are an interesting combination of a solanidine or demissidine aglycone and a lycotetraose or commertetraose sugar moiety. Solanidine forms the aglycone backbone of α-solanine and α-chaconine from potato, while the lycotetraose decoration is found on α-tomatine from tomato (*Distl and Wink, 2009*; *Cárdenas et al., 2015*). The biosynthesis pathways leading to the production of these major SGAs from cultivated potato and tomato have largely been elucidated in recent years and it was found that the underlying genes occur in conserved clusters (*Itkin et al., 2013*; *Cárdenas et al., 2015*). This knowledge and the similarities between SGAs from *S. commersonii* and cultivated potato/tomato will help to identify the missing genes from the pathway through comparative genomics.

The broad-spectrum activity of tetraose SGAs is attractive, but this non-specificity also presents a risk. The antifungal and anti-insect activity of SGAs from *S. commersonii* is not restricted to potato pathogens and pests, but could also affect beneficial or commensal micro-organisms or other animals that feed on plants (*Roddick, 1996*; *Eich, 2008*). In tomato fruit, α-tomatine is converted to esculeoside A during fruit ripening in a natural detoxification process from the plant (*Nakayasu et al., 2020*; *Szymański et al., 2020*) to facilitate dispersal of the seeds by foraging animals. Unintended toxic effects of SGAs should also be taken into account when used in resistance breeding.

Studies on α-tomatine and avenacin A-1 show that changes to the sugar moiety of these saponins from tomato and oat, respectively, can affect their toxicity (*You and van Kan, 2021*; *Roddick, 1974*; *Campbell and Duffey, 1979*; *Sandrock and Vanetten, 1998*). Tomato and oat pathogens produce enzymes that can detoxify these compounds through removal of one or more glycosyl groups (*You and van Kan, 2021*; *Kaup et al., 2005*; *Seipke and Loria, 2008*; *Ökmen et al., 2013*; *Osbourn et al., 1995*; *Bowyer et al., 1995*). The degradation products of saponins can also suppress plant defence responses (*Ito et al., 2004*; *Bouarab et al., 2002*). Conversely, here we show that the resistance of *S. commersonii* is based on the addition of a glycosyl group to a triose saponin from *S. commersonii*. There is large variation in both the aglycone and the sugar moiety of SGAs from wild *Solanum*, with likely over 100 distinct SGAs produced in tubers (*Distl and Wink, 2009*; *Shakya and Navarre, 2008*). This diversity suggests a pressure to evolve novel molecules, possibly to resist detoxification or other tolerance mechanisms, reminiscent of the molecular arms race that drives the evolution of plant immune receptors (*Jones and Dangl, 2006*). Thus, wild *Solanum* germplasm is not only a rich source of immune receptors, it also provides a promising source of natural defence molecules. Studies of how pathogens that naturally occur on *S. commersonii*, or other *Solanum* species producing tetraose SGAs, can tolerate SGAs produced by their hosts could help judge the durability of this type of resistance.

As crops are usually affected by multiple diseases and pests, significant reduction of pesticide use can only be achieved if plants are naturally protected against a range of pathogen species and pests. Different strategies towards this goal have been proposed and our study underlines the potential of

defence compounds that are naturally produced by plants. The fact that genes for specialised plant metabolites can occur in biosynthetic gene clusters (*Itkin et al., 2013*; *Qi et al., 2004*; *Nützmann and Osbourn, 2014*; *Nützmann et al., 2016*), means that introgression breeding could help to move these compounds from wild relatives to crop species. We had already created *S. commersonii* × *S. tuberosum* hybrids with resistance to early blight in a previous study (*Wolters et al., 2021*), but it is clear that potential negative effects of SGA variants on human health and the environment should be considered before these can be developed into a cultivar.

Additional insight in the biosynthesis pathway of the tetraose SGAs produced by *S. commersonii* would make it possible to employ them through metabolic engineering and allow for a more precise control of the amounts that are produced and in which tissues (*Polturak and Osbourn, 2021*). Alternatively, the defence compounds could be produced in non-crop plants or other organisms and applied on crops as biological protectants. Studies on how natural defence compounds are produced in different plant tissues, their toxicity and how they are detoxified, combined with studies on how different modifications ultimately affect plant immunity and toxicity, are essential to employ them in a safe and effective manner. Such studies at the interface of plant immunity and metabolism can help to design innovative solutions to complement existing resistance breeding strategies and improve sustainability of our food production.

## Materials and methods

**Key resources table**

| Reagent type (species) or resource | Designation | Source or reference | Identifiers | Additional information |
|---|---|---|---|---|
| Gene (*Solanum commersonii*) | *ScGTR1* | This paper, sequence deposited at GenBank | GenBank: OM830430 | |
| Gene (*S. commersonii*) | *ScGTR2* | This paper, sequence deposited at GenBank | GenBank: OM830431 | |
| Strain, strain background (*S. commersonii*) | CGN18024 | CGN WUR | CGN18024 | |
| Strain, strain background (*Alternaria solani*) | altNL03003 | *Wolters et al., 2019*; CBS-KNAW | altNL03003; CBS 143772 | |
| Genetic reagent (*S. commersonii*) | CGN18024_3-*ScGTR1*-1,2 and 3 | This paper | | Maintained at plant breeding, WUR; available upon reasonable request |
| Genetic reagent (*S. commersonii*) | CGN18024_3-*ScGTR2*-6,7 and 9 | This paper | | Maintained at plant breeding, WUR; available upon reasonable request |

BSR-Seq was carried out as described in *Dobnik et al., 2021* and *A. solani* disease test were performed following *Wolters et al., 2019*.

### Plant material

Seeds from *S. commersonii* and *S. malmeanum* accessions (*Supplementary file 1*) were obtained from the CGN germplasm collection (Wageningen, the Netherlands). Seeds were sterilised by washing them in 70% ethanol, followed by a 15-min incubation in a 1.2% sodium hypochlorite solution. Sterilised seeds were rinsed three times in sterile tap water and sown out on MS20 medium (4.4 g Murashige and Skoog basal salt mixture including vitamins, 20 g sucrose and 8 g/l micro agar, pH = 5.8) (*Savary et al., 2019*) and incubated in the dark until germinated.

The mapping population was generated by crossing *S. commersonii* CGN18024_1 with CGN18024_3 and vice versa. Ripe berries were harvested about 6 weeks after pollination. Seeds were harvested from the ripe berries, washed with tap water and dried at room temperature on filter paper for 2 weeks. Dry seeds were stored at 4°C until use.

All plants were maintained in tissue culture on MS20 medium. Fresh shoots were propagated 2 weeks prior to transferring plants to soil. Plants were grown in a greenhouse under long-day conditions (16 hr light/8 hr dark).

### Isolation of nucleic acids and sequencing

RNA isolations were performed from leaf material that was harvested from fully expanded leaves of 3-week-old CGN18024_1 and CGN18024_3 and from young leaves from the top of the plant of

3-week-old progeny derived from the cross between these genotypes and of transformants. RNA was extracted using the RNeasy Plant Mini Kit following the manufacturer's instructions, including an on-column DNase treatment (QIAGEN). RNA sequencing was performed on the Illumina platform (PE150) by Novogene (United Kingdom), using around 4 µg of RNA.

Genomic DNA was isolated using the DNeasy Plant Mini Kit (QIAGEN) or in a 96-well format (*Dobnik et al., 2021*). High-molecular-weight genomic DNA was isolated from young leaves of CGN18024_1 and CGN18024_3 as described previously (*Hoopes et al., 2022*; *Bernatzky and Tanksley, 1986*). Quality and integrity of RNA and DNA samples were assessed using nanodrop, Qubit, and gel electrophoresis. ONT sequencing was performed on a Nanopore GridION system, using three flow cells, using about 1 µg of DNA per flow cell and a run-time of 72 hr. Approximately 4 µg of genomic DNA was sent to BGI Europe for sequencing on the DNBseq platform.

## Genome assembly and separation of haplotypes covering resistance region

ONT reads were filtered using Filtlong v0.2.0 (https://github.com/rrwick/Filtlong; *Wick, 2018*) with --min_length 1000 and --keep_percent 90. Adapter sequences were removed using Porechop (*Wick et al., 2017*). Fastq files were converted to Fasta using seqtk v1.3 (https://github.com/lh3/seqtk; *Li, 2018*). Assembly was performed with smartdenovo (https://github.com/ruanjue/smartdenovo/; *Liu et al., 2021*) and a k-mer size of 17, with the option for generating a consensus sequence enabled. ONT reads were mapped back to the assembly using minimap2 v2.17 (*Li and Birol, 2018*) and used for polishing with racon v1.4.3 (*Vaser et al., 2017*) using default settings. DNBseq reads were mapped to the resulting sequence using bwa mem v0.7.17 (*Li, 2013*) and used for a second round of polishing with racon v1.4.3. This procedure to polish the assembly using DNBseq reads was repeated once. ONT reads were mapped back to the polished CGN18024_1 assembly using minimap2 v2.17 (*Li and Birol, 2018*). The alignment was inspected using IGV v2.6.3 (*Robinson et al., 2011*) to identify polymorphisms for new markers and marker information was used to identify ONT reads representative for both haplotypes spanning the resistance region of CGN18024_1. Bedtools v2.25.0 (*Quinlan and Hall, 2010*) was used extract the resistance region from the reads and to mask the corresponding region from the original CGN18024_1 assembly. The extracted resistance regions from both reads were appended to the assembly and the polishing procedure described above was repeated to prepare a polished genome assembly of CGN18024_1, containing a sequence of both haplotypes covering the resistance region. Quality of the genome was assessed using quast v5.0.2 with `--eukaryote --large` (*Gurevich et al., 2013*).

## Comparing haplotypes covering resistance region

Genes were predicted using the funannotate v1.7.4 (https://github.com/nextgenusfs/funannotate/; *Palmer and Stajich, 2020*) pipeline. Briefly, funannotate was used to sort and mask the genome and training was performed using the BSA-RNAseq data (`--max_intronlen 10000`). Gene prediction was prepared using the `--optimize_augustus --organism other` and `--max_intronlen 10000` options. The two haplotypes covering the resistance region were compared using nucmer and visualised using mummerplot from the MUMmer4 package (*Marçais et al., 2018*).

## Development of markers and genotyping

Bedtools v2.25.0 (*Quinlan and Hall, 2010*) was used to extract the regions surrounding polymorphisms from the DMv4.03, Solyntus and CGN18024_1 genomes. Primers were designed using Batch-Primer3 (*You et al., 2008*; *Supplementary file 5*). HRM markers were amplified with Phire Hot Start II DNA Polymerase (Thermo Fisher Scientific) and genotyped on a LightScanner System (Bio Fire) as described previously (*Dobnik et al., 2021*). InDel markers were amplified using DreamTaq DNA Polymerase (Thermo Fisher Scientific) and visualised using gel electrophoresis following standard laboratory protocols.

## Cloning of candidate resistance genes

*ScGTR1*, *ScGTR2*, and *ScGTS* were amplified from genomic DNA from CGN18024_1 using Phusion Polymerase (New England BioLabs) and the primers listed in *Supplementary file 6* following standard laboratory protocols. cacc was included at the 5' end of each forward primer to facilitate cloning in

the pENTR D-TOPO vector (Thermo Fisher Scientific), following the manufacturer's instructions. Insert sequences were validated through Sanger sequencing (Macrogen Europe). The genes were cloned into the pK7WG2 vector (*Karimi et al., 2002*) using Gateway LR Clonase II (Thermo Fisher Scientific) following the manufacturer's instructions and transformed to electrocompetent *A. tumefaciens* AGL1 (*Lazo et al., 1991*) containing the helper plasmid pVirG (*van der Fits et al., 2000*).

## Transient disease assay

Agroinfiltration was performed as described previously using *Agrobacterium tumefaciens* strain AGL1 (*Lazo et al., 1991*; *Domazakis et al., 2017*). *Agrobacterium* suspensions were used at an $OD_{600}$ of 0.3 to infiltrate fully expanded leaves of 3-week-old CGN18024_1, CGN18024_3 and *S. tuberosum* cv. Atlantic. pK7WG2-*ScGTR1*, pK7WG2-*ScGTR2*, pK7WG2-*ScGTS*, and pK7WG2-empty were combined as four separate spots on the same leaf and the infiltrated areas were encircled with permanent marker. The plants were transferred to a climate cell 48 hr after agroinfiltration and each infiltrated area was inoculated with *A. solani* by pipetting a 10-μl droplet of spore suspension ($1 \times 10^5$ conidia/ml) at the centre of each spot. Lesion diameters were measured 5 days post inoculation. Eight plants were tested of each genotype, using three leaves per plant.

## Fungal growth inhibition assays

Mature leaf material from 5-week-old plants was extracted in phosphate-buffered saline buffer using a T25 Ultra Turrax disperser (IKA) and supplemented to obtain a 5% (wt/vol) suspension in PDA and autoclaved (20 min at 121°C), or added to PDA after autoclaving, followed by an incubation step for 15 min at 60°C to semi-sterilise the medium. The medium was poured into Petri dishes. Small agar plugs containing mycelium from *A. solani* (altNL03003) or *F. solani* (1992 vr) were placed at the centre of each plate and the plates were incubated at 25°C in the dark. Similarly, approximately 100 spores of *B. cinerea* B05.10 (*Amselem et al., 2011*) were pipetted at the centre of PDA plates containing the different leaf extracts and the plates were incubated at room temperature in the dark. Three plates per fungal isolate/leaf extract combination were prepared and colony diameters were measured daily using a digital calliper.

## Potato transformation

Internodes from in vitro grown plants were used to generate stable transformants using previously described methods (*Hoekema et al., 1989*; *Fillatti et al., 1987*). Transformants were selected on MS20 containing 100 μg/ml kanamycin. Successful transformants were characterised using primers listed in *Supplementary file 7*.

## SGA measurements

Mature leaves were harvested from three different 5-week-old plants of each genotype in 2 ml tubes containing two steel beads and flash frozen in liquid nitrogen. Leaf material was ground using a TissueLyser II bead mill (QIAGEN). Approximately 100 mg of ground leaf material was extracted in 1 ml of 70% methanol and 0.1% formic acid. Samples were vortexed and sonicated for 15 min in an Ultrasonic Cleaner (VWR). Samples were vortexed once more and centrifuged for 15 min in a tabletop centrifuge at $17,000 \times g$. The supernatant was passed through 0.45 μm syringe filters (BGB) and diluted 5×. The extracts were separated on Acquity UPLC HSS T3 1.8 μm (2.1 × 150 mm) column Acquity UPLC H Class Plus system (Waters). The separation was performed using the following water + 0.1 formic acid/acetonitrile + 0.1% formic acid gradient: initial – A/B = 95/5%, 65/35% – 14 min, 55/45% – 20 min, 15/85% – 24 min, 95/5% – 25 min, 95/5% – 30 min. The MS data were acquired using an Acquity QDa mass spectrometer (Waters) in negative and positive mode (in separate runs) from 150 to 1250 Da, cone voltage 15 V, capillary voltage 0.8 kV at 2 scans/s acquisition rate. The raw chromatograms were subjected to full spectra alignment using Metalign software (https://www.wur.nl/en/show/MetAlign-1.htm). SGAs were putatively identified using MS fragmentation patterns (*Supplementary files 3 and 4*), which were compared with MS information available in literature (*Osman et al., 1976*; *Distl and Wink, 2009*; *Caruso et al., 2011*; *Vázquez et al., 1997*).

## CPB test

CPB was reared on *S. tuberosum* cultivar 'Bintje' in insect rearing cages in a greenhouse compartment at 25/23°C and 16/8 hr light/dark photoperiod and 70% relative humidity. Freshly laid egg packages were removed from the rearing every day and, once hatched, 1-day-old larvae were used for the experiment. Resistance to CBP was measured by assessing mortality and weight of CPB larvae on three plants of each genotype in a non-choice assay. At the start of the experiment, five 1-day-old larvae were placed in a clip-cage on a leaf and an insect sleeve was used to enclose every plant to restrict the larvae to the plant. Larvae were able to feed for 9 days, after which surviving larvae were counted and weighed on a scale. If less than five larvae were found on the plant, the remainder was assumed dead.

## Data analysis

Data were analysed in RStudio (R version 4.02) (*RStudio Team, 2020*; *R Development Core Team, 2020*), using the tidyverse package (*Wickham et al., 2019*). Most figures were generated using ggplot2 (*Wickham, 2016*), but genomic data were visualised using Gviz and Bioconductor (*Hahne and Ivanek, 2016*). PCA was performed using PAST3 software (https://past.en.lo4d.com/windows). p values for comparisons between means of different groups were calculated in R using Welch's two-sample *t*-test. Experimental replicates are from biological distinct samples. Experiments were repeated at least twice with similar results.

## Data availability

RNAseq data from the BSR-Seq experiment were deposited in the NCBI Sequence Read Archive with BioProject ID PRJNA792513 (Sequencing Read Archive accession IDs SRR17334110, SRR17334111, SRR17334112, and SRR17334113). Raw reads used in the assembly of the CGN18024_1 genome were deposited with BioProject ID PRJNA789120 (Sequencing Read Archive accession IDs SRR17348659 and SRR17348660). The assembled genome sequence of CGN18024_1 was archived on NCBI as WGS project JAJTWQ01 (GenBank assembly GCA_029007595.1). Sequences of *ScGTR1* and *ScGTR2* were deposited in GenBank under accession numbers OM830430 and OM830431. Numerical data underlying the figures of this manuscript are included as source data files.

## Acknowledgements

This research was funded by the J.R. Simplot Company, we especially thank Craig Richael for his support and useful discussions. We thank Dirk Jan Huigen and Henk Meurs for taking care of the plants in the greenhouse and Jack Vossen for providing us with *F. solani* isolate 1992 vr from the collection of Biointeractions and Plant Health (Theo van der Lee, WUR). Jan van Kan and Yaohua You for insightful discussions and for providing us with *B. cinerea* isolate B05.10. Evert Jacobsen for his feedback on the manuscript. Martijn van Kaauwen and Richard Finkers for bioinformatics support. PJW thanks Andrea Lorena Herrera for her support and helpful talks about specialised plant metabolites.

## Additional information

### Competing interests

Pieter J Wolters, Richard GF Visser, Vivianne GAA Vleeshouwers: PJW, RGFV and VGAAV are inventors on U.S. Patent Application No. 63/211,154 relating to ScGTR1 and ScGTR2 filed by the J.R. Simplot company. The other authors declare that no competing interests exist.

### Funding

| Funder | Grant reference number | Author |
| --- | --- | --- |
| J.R. Simplot Company | | Pieter J Wolters<br>Doret Wouters<br>Vivianne GAA Vleeshouwers |

| Funder | Grant reference number | Author |
|---|---|---|

The funders had no role in study design, data collection, and interpretation, or the decision to submit the work for publication.

## Author contributions

Pieter J Wolters, Conceptualization, Resources, Data curation, Formal analysis, Supervision, Funding acquisition, Validation, Investigation, Visualization, Methodology, Writing – original draft, Project administration, Writing – review and editing; Doret Wouters, Formal analysis, Supervision, Validation, Investigation, Methodology, Project administration; Yury M Tikunov, Formal analysis, Investigation, Methodology, Writing – review and editing; Shimlal Ayilalath, Formal analysis, Investigation; Linda P Kodde, Miriam F Strijker, Methodology; Lotte Caarls, Formal analysis, Supervision, Methodology; Richard GF Visser, Conceptualization, Funding acquisition, Project administration, Writing – review and editing; Vivianne GAA Vleeshouwers, Conceptualization, Resources, Supervision, Funding acquisition, Investigation, Writing – original draft, Project administration, Writing – review and editing

## Author ORCIDs

Pieter J Wolters https://orcid.org/0000-0001-5941-9189

Reviewer #1 (Public Review): https://doi.org/10.7554/eLife.87135.3.sa1
Reviewer #2 (Public Review): https://doi.org/10.7554/eLife.87135.3.sa2
Author Response https://doi.org/10.7554/eLife.87135.3.sa3

---

# Additional files

## Supplementary files

• Supplementary file 1. *Solanum commersonii* and *Solanum malmeanum* accessions used in this study. Accessions were obtained from the Centre for Genetic Resources, the Netherlands (CGN WUR). Two to three genotypes from each accession were used in the disease screen with *A. solani*.

• Supplementary file 2. Overview of characterised glycosyltransferases (GTs) used in comparative phylogenetic analysis (*Figure 2—figure supplement 8*). GTs with a known function are taken from *Bowles et al., 2005*, *McCue et al., 2005*, *McCue et al., 2006*, *McCue et al., 2007*, *Masada et al., 2009*, *Itkin et al., 2013*, *Itkin et al., 2011*, and *Tikunov et al., 2013*.

• Supplementary file 3. Putative identities and relative contents of steroidal glycoalkaloids (SGAs) in different potato genotypes. Average signal intensities (3 replicates per genotype) are presented as a percentage of the maximum signal intensity.

• Supplementary file 4. Overview of the steroidal glycoalkaloids detected in our study. RT – retention time; [M−H+FA]− – mass of a molecular ion at negative ionisation mode (all alkaloids were represented by formic acid adduct ions); [M+H]+ – mass of a molecular ion at negative ionisation mode; Putative structure – putative combination of aglycones and sugar moieties deduced by comparing the fragmentation spectrum derived at positive ionisation with previous studies (*Osman et al., 1976*; *Distl and Wink, 2009*; *Caruso et al., 2011*; *Vázquez et al., 1997*); Fragmentation spectra derived using positive ionisation: P – parent ion or P-fragment(s) loss.

• Supplementary file 5. Primers used to map the resistance region.

• Supplementary file 6. Primers used to clone candidate resistance genes.

• Supplementary file 7. Primers used to validate transformants.

• MDAR checklist

## Data availability

RNAseq data from the BSR-Seq experiment were deposited in the NCBI Sequence Read Archive with BioProject ID PRJNA792513 (Sequencing Read Archive accession IDs SRR17334110, SRR17334111, SRR17334112, and SRR17334113). Raw reads used in the assembly of the CGN18024_1 genome were deposited with BioProject ID PRJNA789120 (Sequencing Read Archive accession IDs SRR17348659 and SRR17348660). The assembled genome sequence of CGN18024_1 was archived on NCBI as WGS project JAJTWQ01 (GenBank assembly GCA_029007595.1). Sequences of ScGTR1 and ScGTR2 were

deposited in GenBank under accession numbers OM830430 and OM830431. Numerical data underlying the figures of this manuscript are included as Figure source data files.

The following datasets were generated:

| Author(s) | Year | Dataset title | Dataset URL | Database and Identifier |
|---|---|---|---|---|
| Wolters PJ, Wouters D, Tikunov YM, Ayilalath S, Kodde L, Strijker M, Caarls L, Visser RGF, VGAA Vleeshouwers | 2023 | BSR-Seq data | https://www.ncbi.nlm.nih.gov/sra/?term=PRJNA792513 | BioProject, PRJNA792513 |
| Wolters PJ, Wouters D, Tikunov YM, Ayilalath S, Kodde L, Strijker M, Caarls L, Visser RGF, VGAA Vleeshouwers | 2023 | Solanum commersonii isolate:CGN18024_1 (Commerson's wild potato) | https://www.ncbi.nlm.nih.gov/bioproject/?term=PRJNA789120 | BioProject, PRJNA789120 |
| Wolters PJ, Wouters D, Tikunov YM, Ayilalath S, Kodde L, Strijker M, Caarls L, Visser RGF, VGAA Vleeshouwers | 2023 | Genome assembly CGN18024_1v5_2 | https://www.ncbi.nlm.nih.gov/datasets/genome/GCA_029007595.1/ | GenBank assembly, GCA_029007595.1 |
| Wolters PJ, Wouters D, Tikunov YM, Ayilalath S, Kodde L, Strijker M, Caarls L, Visser RGF, VGAA Vleeshouwers | 2023 | Solanum commersonii glucosyltransferase (GTR1) mRNA, complete cds | https://www.ncbi.nlm.nih.gov/nuccore/OM830430 | NCBI GenBank, OM830430 |
| Wolters PJ, Wouters D, Tikunov YM, Ayilalath S, Kodde L, Strijker M, Caarls L, Visser RGF, VGAA Vleeshouwers | 2023 | Solanum commersonii xylosyltransferase (GTR2) mRNA, complete cds | https://www.ncbi.nlm.nih.gov/nuccore/OM830431 | NCBI GenBank, OM830431 |

The following previously published datasets were used:

| Author(s) | Year | Dataset title | Dataset URL | Database and Identifier |
|---|---|---|---|---|
| Xu X, Pan S, Cheng S, Zhang B, Mu D, Ni P | 2011 | DMv4.03 genome | http://spuddb.uga.edu/pgsc_download.shtml | Spud DB, PGSC v4.03 |
| van Lieshout N, van der Burgt A, de Vries ME, ter Maat M, Eickholt D, Esselink D | 2020 | Solyntus v1.1 genome assembly | https://www.plantbreeding.wur.nl/Solyntus/ | Solyntus Genome Sequencing Consortium, v1.1 |

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
