## [Editor Report · eLife assessment]

This **valuable** study links natural variation in steroidal glycoalkaloid production to disease and insect resistance in potato species. The study design is straightforward and thorough, and the evidence supporting the main conclusions is **solid**. The work will be of interest to plant biologists and breeders.

---

## [Referee Report · Reviewer #1 (Public Review)]

This manuscript conducts a classic QTL analysis to identify the molecular basis of natural variation in disease resistance. This identifies a pair of glycosyltransferases that contribute to steroidal glycoalkaloid production. Specifically altering the final hexose structure of the compound. This is somewhat similar to the work in tomatine showing that the specific hexose structure mediates the final potential bioactivity. Using the resulting transgenic complementation lines that show that the gene leads to a strong resistance phenotype to one isolate of Alternaria solani and the Colorado potato beetle. This is solid work showing the identification of a new gene and compound influencing plant biotic interactions. The authors have improved the introduction and discussion to better show the breadth of knowledge in pathogen-defense metabolite interactions involving plants.

---

## [Referee Report · Reviewer #2 (Public Review)]

The study focuses on a mechanism of pest/pathogen resistance identified in Solanum commersonii, which appears to offer dominant resistance to Alternaria solani (potato early blight) through the activity of specific glycosyltransferases which facilitate the production of tetraose glycoalkaloids in leaf tissue. The authors demonstrated that these glycoalkaloids are suppressive to the growth of multiple pathogenic ascomycetes and furthermore, that transgenic plants expressing these glycosyltransferases in susceptible S. commersonii clones demonstrate improved resistance to specific strains of A. solani and a genotype of Colorado Potato Beetle. The study design is straightforward, yet thorough, and does a good job demonstrating the importance of these genes in resistance. This work is significant because it demonstrates the mechanism behind resistance to a necrotrophic pathogen. Resistance to this group of pathogens has historically relied on mechanisms that do not include the use of typical dominant resistance gene products (nucleotide-binding, leucine-rich repeat proteins). The identification of these glycosyltransferases and their role in resistance will give potato breeders options for the development of markers associated with resistance to this group of pathogens. However, this may demonstrate an important battle to balance between production traits (like disease resistance) and quality traits (like glycoalkaloid content), as the two may be mutually exclusive in the development of new varieties.

---

## [Author Response]

The following is the authors’ response to the original reviews.

Both reviewers strongly suggest that you modify the title of your paper to something that better reflects the data presented.

We have made the title more specific to the findings described in the manuscript and revised the rest of the manuscript in response to the additional reviewer’s comments. We adjusted the abstract accordingly.

**Public Reviews:**

**Reviewer #1 (Public Review):**
This manuscript conducts a classic QTL analysis to identify the molecular basis of natural variation in disease resistance. This identifies a pair of glycosyltransferases that contribute to steroidal glycoalkaloid production. Specifically altering the final hexose structure of the compound. This is somewhat similar to the work in tomatine showing that the specific hexose structure mediates the final potential bioactivity. Using the resulting transgenic complementation lines that show that the gene leads to a strong resistance phenotype to one isolate of Alternaria solani and the Colorado potato beetle. This is solid work showing the identification of a new gene and compound influencing plant biotic interactions. While the experiments are solid, the introduction, discussion and associated claims don't accurately reflect my reading of what is known and said in the current literature.The sentence on line 53-54 is misleading. It provides only three citations on specific links between specialized metabolism and disease resistance. However, there are actually at least 40 on specific links of camalexin and indolic phytoalexins to disease resistance. Similarly there are dozens of uncited papers on benzoxazinoids, indolic glucosinolates, aliphatic glucosinolates and tomatine to both non-host and host based resistance mechanisms. This even goes as far as showing how the pathogens resist an array of these compounds. The choices in the introduction make it appear that little is known about specialized metabolism to disease resistance but I would suggest that this is not an allusion supported by the literature. I would agree that given the breadth of specialized metabolism we have a lot of knowledge about a set of them but that there are hundreds to thousands of untested compounds but to indicate that little is known is unfair to the specialized metabolism community. This is especially true as the introduction and discussion give no image of the large body of literature on specialized metabolism to insect interactions even though this is a major component of this manuscript.

We have rewritten this part of the introduction (lines 50-69). In the original text, we meant to convey our impression that receptor-mediated resistance is studied in a very high degree of detail, and that resistance that is based on secondary metabolites is receiving less recognition in comparison, especially in the plant-microbe interactions field. We agree that our comments might give the (false) impression that there is not much known. There is indeed a lot of data to support the importance of specialised metabolites in resistance, especially against necrotrophic pathogens and insects. The changes that we made should give a better reflection of that knowledge.

I would also agree that specialized metabolism is not a conscious target of breeding programs but the work on benzoxazinoids in maize and glucosinolates in the Brassica's has shown that these compounds have been influenced by breeding programs. Similarly work on de novo domestication of multiple crops is focused on the adjustment of specialized metabolism in these crops.

The reviewer is right to point out that specialized metabolism is influenced by breeding. Specialized metabolites may not only be involved in defence, but they can also affect other properties of the plant such as quality aspects. Potato breeders have made efforts to reduce SGA content in tubers to prevent problems with toxicity and to meet safety regulations. We have adjusted the discussion (lines 255-260).

I would disagree with the hint on line 49-50 and again on lines 236-239 that specialized metabolism may have less pleiotropy. This is not supported by recent work on benzoxazinoids and glucosinolates showing that they have numerous regulatory links to the plant and can be highly pleiotropic. Even the earliest avenicin work in oat showed that the deficient lines had altered root development.

We agree with the reviewer and we have removed the hints that specialized metabolism may have less pleiotropy from the manuscript. We do believe that the broad-spectrum activity of specialized metabolites can be an advantage, but this non-specificity also comes with risks in case of food crops. We note the potential negative effects of SGAs in the discussion (see previous comment and lines 300-303).

My main message from the above three paragraphs is to point out that there are a number of places in the manuscript where the current state of the specialized metabolite literature is not accurately portrayed. To properly place the manuscript in the broader context, I would suggest a more even handed introduction and discussion that takes into account the current state of the specialized metabolism literature.

We rewrote these parts to provide a more balanced view on the role of specialized metabolites in disease resistance.

Is it accurate to say complete resistance to A. solani if only a single isolate of the pathogen is used? Is there evidence that I am unaware of that there are no isolates of this pathogen with saponin resistance? There are pathogens with natural tomatine resistance and this is a common feature of plant pathogens that they have genetic variation in the resistance to specialized metabolism. For example, it should be noted that Botrytis BO5.10 is a tomatine sensitive isolate and the van Kan and Hahn groups have published on isolates that are resistant to saponins. I would suggest caveating across the manuscript that this is a single isolate and that it is possible that there may be isolates with natural resistance to the steroidal glycoalkaloid?

While it is true that we only describe the results of testing a single isolate of A. solani in the submitted manuscript, we previously showed that the S. commersonii resistance is effective against additional Alternaria isolates and species from different locations (1). We included this context to the introduction (lines 71-73) and also added the results of testing a more recent Dutch A. solani isolate (altNL21002, isolated from a potato field in the Netherlands in 2021) and an isolate from the US (ConR1H, isolated from a potato field in Idaho in 2015) to the supplementary material of the revised manuscript (lines 102-104). Of course, this still does not prove that the SGAs protect against all A. solani isolates and we have been more specific in referring to the Alternaria isolate that was tested.Similarly, it is impossible to make a general statement on the lack of detoxification capacity of all isolates of A. solani. It may indeed be possible that there are Alternaria isolates that are tolerant to the tetraose SGAs produced by S. commersonii, especially in natural habitats where Solanum species that produce tetraose SGAs and Alternaria co-occur. We have added this point to the discussion (lines 292-294).

In Figure 4b, is the infection site about 3.5 mm in size such that 3.5 mm means absolutely no infection? If not, that would mean there is some outgrowth by Alternaria and the resistance isn’t complete.

We often observe dead tissue underneath the inoculation droplet on resistant plants, which is measured as a lesion. Such lesions can usually visually be discriminated from the lesions on susceptible genotypes by their colour (dark black for resistant plants versus a more brownish colour of the lesions on susceptible plants), but this information is lost in the quantitative data presented in the figures. Droplets occasionally flow out over the leaf surface, which may explain why larger ‘lesions’ are sometimes observed on resistant plants. In rare cases, there may also be a little bit of outgrowth of Alternaria beyond the inoculation droplet before the infection is stopped on resistant genotypes. Whether the resistance is ‘complete’ in such cases is debatable. We tuned down our statements regarding ‘complete’ resistance throughout the manuscript.

**Reviewer #2 (Public Review):**
The study focuses on a mechanism of pest/pathogen resistance identified in Solanum commersonii, which appears to offer dominant resistance to Alternaria solani through the activity of specific glycosyltransferases which facilitate the production of tetraose glycoalkaloids in leaf tissue. The authors demonstrated that these glycoalkaloids are suppressive to the growth of multiple pathogenic ascomycetes and furthermore, that transgenic plants expressing these glycosyltransferases in susceptible S. commersonii clones demonstrate improved resistance to a specific strain of A. solani and a genotype of Colorado Potato Beetle. The study design is straightforward, yet thorough, and does a good job demonstrating the importance of these genes in resistance. While the research findings are significant there are statements throughout the manuscript that overstate both the novelty and utility of the findings.Title: While the protection is impressive, the title suggests that these glycoalkaloids provide protection against all fungi and insects, which is both unlikely and essentially impossible to prove. This should be changed to something more measured. This is especially true given that only a single fungus and insect were tested against transgenic plants, but would be an overstatement even with more robust evaluation.

We appreciate the comment of the reviewer and agree that is unlikely that the S. commersonii SGAs protect against all fungi and insects and that it would be impossible to prove this. We intended to highlight the fact that these compounds provide a qualitative (‘complete’) resistance against the tested isolates/genotypes, and that they are effective across a wide range of organisms (‘fungi and insects’). We have made the title more specific to the findings described in the manuscript.

Throughout the paper: A single isolate of A. solani and a single genotype of CPB were used in this study. While this is in line with the typical limitations of such a study, the authors need to be careful about claiming broad resistance to either of the species. Variability in fungicide tolerance and detoxification activity have been noted in both fungi and CPB, so more specific language should be used throughout (such as L213 and L221).

Similar points were raised by reviewer 1. We have tuned down our statements regarding ‘complete’ resistance and clarified that we tested only a limited set of A. solani isolates and single CPB genotype throughout the manuscript.

**Reviewer #2 (Recommendations For The Authors):**
L39: Fix grammar.

Done

L42: Race is a terminology not used in all pathosystems (others include pathovar, subspecies, etc.).

We removed the word race and use the general ‘pathogen’.

L53: The role of pterocarpans, flavonoids, indoles, terpenes, and a number of other compound classes have been linked to plant defense across the entire plant kingdom. Highlighting Avenacin is fine, but it shouldn't be ignored that the role of phytoalexins and phytoanticipins in defense against fungi (and the subsequent detoxification of these compounds by fungi) has been well established in a number of pathosystems.

We have removed the specific reference to avenacin (we still refer to it in the discussion, as there are interesting similarities with the saponins from tomato and potato) and tried to highlight the diversity of plant defence compounds across the plant kingdom and the importance of tolerance mechanisms in different pathosystems in the revised manuscript (lines 52-60).

L234-237: This is broadly an overstatement. To my knowledge there is quite a bit of interest in plant defense compounds for breeding (in plants generally) and we know quite a bit about their mode of action (fungal membrane perturbation through binding to ergosterol). There have been active breeding efforts for decades to reduce glycoalkaloid content in potatoes due to the hemolytic activity of these compounds. While this may or may not be the case with these specific SGAs, a more accurate summary of the state of the field is warranted.

We have rewritten the paragraph to give a more balanced view of breeding for SGAs in potato (lines 63-69 on the mode of action of SGAs and lines 255-260 regarding breeding for specific SGA variants in potato).

L279: "...introgression breeding could help to move these compounds from wild relatives to crop species..." Yes, but at what cost? If it results in increase GSAs in tubers, then the plants would be inedible. This could be made more clear and support the following statement that alternative deployment techniques including application as biological protectants.

The reviewer is right to point out the importance of considering negative effects of SGAs in breeding. We paid more attention to this aspect in the discussion and added a sentence to clarify that effects on human health and the environment should be considered before employing these compounds (lines 300-303).

Discussion:L229-230: the authors state that the tetraose SGA from commersonii can protect against other fungi, but this does not appear to have been tested. Rather, they looked at resistance in the CGN18024_1 and CGN18024_3 lines, which could express other factors unrelated to GSAs to impact resistance or susceptibility. Experiments to support this statement would include screening of the transgenic lines for resistance to other fungi, but this does not appear to have been done.

We believe that the tetraose SGAs have the potential to protect against a range of fungi, but the reviewer correctly points out that these experiments do not provide definitive proof for their role in resistance to other pathogens besides A. solani and CPB. We have adjusted our statement accordingly (lines 247-250 of the discussion, 84-88 of the introduction and the abstract).

Future questions should likely include characterizing the overall SGA content of resistant potatoes, characterizing the saponin content specifically found within tubers, and purifying the compounds to characterize the hemolytic activity of these specific compounds. Even if these aren't your exact plans, they would be necessary steps in any resistance breeding efforts. In particular, it will be important to know if the SGA content is increased in tubers of the tested lines, especially CGN18024_1, CGN18024_3, and the transgenics. Ideally, for breeding purposes there would be a disconnect between SGA production in foliage and tubers. It is unclear whether this is possible in these lines.

These are all good questions, and it would be nice to follow up on them in future research. We explore the different routes towards a safe use of SGAs in resistance breeding in the discussion.

It has been shown that commersonine, one of the tetraose glycoalkaloids is also present in Solanum chacoense. It would be useful to note both this fact and that the Early Blight resistance which has been noted in Solanum chacoense may additionally be from these compounds (examples below).o https://www.cabi.org/GARA/FullTextPDF/Pre2000/19871336643.pdfo https://apsjournals.apsnet.org/doi/pdf/10.1094/PHYTO-06-18-0181-R (breeding line 24-24-12 has s. chacoense parentage)o https://agris.fao.org/agris-search/search.do?recordID=DJ20220231195

This is indeed an interesting observation and it is well possible that SGAs are responsible for the resistance of S. chacoense. There are additional wild Solanum species that produce similar SGAs as found in S. commersonii that could confer resistance to early blight (or CPB) and we added this to the discussion (lines 263-265).

Reference

1. Wolters PJ, de Vos L, Bijsterbosch G, Woudenberg JH, Visser RG, van der Linden G, et al. A rapid method to screen wild Solanum for resistance to early blight. European Journal of Plant Pathology. 2019;154:109-14.